# Natural Killer Cells as Key Players of Tumor Progression and Angiogenesis: Old and Novel Tools to Divert Their Pro-Tumor Activities into Potent Anti-Tumor Effects

**DOI:** 10.3390/cancers11040461

**Published:** 2019-04-01

**Authors:** Barbara Bassani, Denisa Baci, Matteo Gallazzi, Alessandro Poggi, Antonino Bruno, Lorenzo Mortara

**Affiliations:** 1Immunology and General Pathology Laboratory, Department of Biotechnology and Life Sciences, University of Insubria, Via Monte Generoso, n. 71, 21100 Varese, Italy; 2School of Medicine and Surgery, University of Milano-Bicocca, 20900 Monza, Italy; denisa.baci@gmail.com; 3Vascular Biology and Angiogenesis Laboratory, Scientific and Technologic Park, IRCCS MultiMedica, 20138 Milan, Italy; 91matteogallazzi@gmail.com (M.G.); 82antonino.bruno@gmail.com (A.B.); 4UOSD Molecular Oncology and Angiogenesis Unit, IRCCS Ospedale Policlinico San Martino, 16132 Genoa, Italy; alessandro.poggi@hsanmartino.it

**Keywords:** NK cells, tumor microenvironment, angiogenesis, tumor therapy, targeting immunotherapy, chemotherapy

## Abstract

Immune cells, as a consequence of their plasticity, can acquire altered phenotype/functions within the tumor microenvironment (TME). Some of these aberrant functions include attenuation of targeting and killing of tumor cells, tolerogenic/immunosuppressive behavior and acquisition of pro-angiogenic activities. Natural killer (NK) cells are effector lymphocytes involved in tumor immunosurveillance. In solid malignancies, tumor-associated NK cells (TANK cells) in peripheral blood and tumor-infiltrating NK (TINK) cells show altered phenotypes and are characterized by either anergy or reduced cytotoxicity. Here, we aim at discussing how NK cells can support tumor progression and how induction of angiogenesis, due to TME stimuli, can be a relevant part on the NK cell-associated tumor supporting activities. We will review and discuss the contribution of the TME in shaping NK cell response favoring cancer progression. We will focus on TME-derived set of factors such as TGF-β, soluble HLA-G, prostaglandin E_2_, adenosine, extracellular vesicles, and miRNAs, which can exhibit a dual function. On one hand, these factors can suppress NK cell-mediated activities but, on the other hand, they can induce a pro-angiogenic polarization in NK cells. Also, we will analyze the impact on cancer progression of the interaction of NK cells with several TME-associated cells, including macrophages, neutrophils, mast cells, cancer-associated fibroblasts, and endothelial cells. Then, we will discuss the most relevant therapeutic approaches aimed at potentiating/restoring NK cell activities against tumors. Finally, supported by the literature revision and our new findings on NK cell pro-angiogenic activities, we uphold NK cells to a key host cellular paradigm in controlling tumor progression and angiogenesis; thus, we should bear in mind NK cells like a TME-associated target for anti-tumor therapeutic approaches.

## 1. Introduction

Strong evidences suggest that the presence of inflammatory cells within the TME plays a crucial role in the development and/or progression of tumors [1,2,3]. Among the host-dependent biological features of the tumor hallmarks defined by Hanahan and Weinberg [4], there are “evading immune destruction” and “tumor-promoting inflammation”, which together with the immune cell-mediated orchestration of angiogenesis, point out the key role of the immune system in neoplastic disease [3,4,5]. As a consequence of their functional plasticity, several immune cells, can modify upon stimuli delivered by the components of TME their phenotypic and functional features; this leads to a reduced killing of tumor cells, the expression of a tolerogenic/immunosuppressive behavior and the acquisition of pro-angiogenic activities, thus promoting tumor expansion [1,3,5,6,7].

NK cells are innate lymphocytes that can potentially control tumor growth by their cytotoxic activity [8,9]. Classical NK cells are distinct from innate lymphoid cells (ILCs) although they share with ILC1 several phenotypic features [10,11,12]; indeed, NK cells are key cytolytic effectors of innate immunity while ILC1 are generally non-cytotoxic or weakly cytotoxic [12] but they show a central role in response to certain infections and are also involved in tissue remodeling homeostasis, morphogenesis, metabolism, repair, and regeneration [10]. According to Vivier et al., ILC and NK cells originate from a common lymphoid progenitor (CLP) [11,12]. GATA3 or TOX/NFIL3/ID2/ETS1 drive the distinction between common innate lymphoid progenitor (CLIP) and the NK cell progenitor (NKP), respectively. Finally, T-bet/EOMES expression in NKPs govern NK cell differentiation [11,12]. Natural killer cell subsets can differ according to tissue distribution that is related to distinct homing properties and/or local maturation [13].

According to the surface expression of CD56 and CD16, two major peripheral blood NK subsets have been identified [8,9]. CD56^dim^CD16^+^ NK cells (90–95% of total circulating NK cells), endowed with cytotoxic activities by perforin and granzyme release and mediating antibody dependent cellular cytotoxicity (ADCC) and CD56^bright^CD16^-^ NK cells (5–10% of total circulating NK cells), able in producing Th1 cytokines, such as IFN-γ and TNF-α [8,9]. Whether CD56^dim^CD16^+^ and CD56^bright^CD16^-^ cell subsets can be definitely considered terminally differentiated NK cells, still represent a matter of debate. Strong evidence supports that CD56^bright^ NK cells represent still an immature phenotype that is able to differentiate in CD56^dim^ NK cells in vitro and in humanized murine models [13,14,15]. A distinct NK cell subset was found within the developing decidua known as decidual NK cells (dNK). dNK cells are able to acquire a tolerogenic and pro-angiogenic phenotype, identified as CD56^superbright^CD16^-^VEGF^high^PlGF^high^CXCL8^+^ dNKs and are necessary to drive the spiral artery formation during the embryo development [16,17]. Alterations of the expression of relevant activating receptors such as the natural cytotoxicity receptors (NCRs: NKp30, NKp44, and NKp46) have been observed in blood from acute myeloid leukemia (AML) patients [18]; in addition, recent studies in breast [19], lung [20,21], colorectal cancer (CRC) [22,23], renal cell carcinoma [24], and gastrointestinal stromal tumors [25] have shown that intratumor NK cells display phenotypic and/or functional alterations compared with peripheral NK cells.

Neoplastic transformation significantly impacts on NK cell phenotype, localization, and functions. CD56^bright^CD16^low/−^Perf^low^ NK cells appears to preferentially accumulate in solid cancers [2,5,20,21,22,26,27,28,29,30]. Recently, a new NK cell subset, termed CD56^low^CD16^low^, has been described in the bone marrow (BM) and peripheral blood of pediatric healthy donors and leukemic transplanted patients. This CD56^low^CD16^low^ NK cell subset is supposed to represent an intermediate stage of differentiation between CD56^high^CD16^+/−^ and CD56^low^CD16^high^ [31,32,33]. Elevated number of CD56^low^CD16^low^ NK cells have also been found in the BM on multiple myeloma patients, with decreased expression of activating receptors such as DNAM-1 and NKp30 and impaired cytolytic capabilities [15].

## 2. TME Factors Orchestrating NK Cell Activity

Tumor cells have developed several mechanisms to evade from NK cell immunosurveillance, through the modulation of cell surface molecules involved in their recognition and the release of immunosuppressive soluble factors such as TGF-β, HLA-G, prostaglandins and adenosine in the TME [34].

### 2.1. Selected Soluble Factors

#### 2.1.1. TGF-β

Suppressive cytokines are crucial orchestrators in shaping NK cell anergy and exhaustion in tumors [35]. TGF-β is a major immunosuppressive cytokine present in the TME [36,37] and it is detected at high levels in different tumors [36]. The inhibitory effects of TGF-β on NK cells are well documented and act mainly by downregulating the expression of NKG2D [38] (Figure 1A). TGF-β has also been shown to inhibit CD16-mediated human NK cell IFN-γ production and ADCC through SMAD3 [39].

NK cells from healthy donors, following exposure to TGF-β, exhibit alteration in their killing capability through inhibition of perforin and granzyme release [40,41,42]. Further, TGF-β-mediated alteration of degranulation capability (i.e., CD107a release) and reduction in Th1 cytokines have been described for NK cells in different tumors [36,37,38]. Mechanistically, Viel et al. demonstrated that the mammalian target of rapamycin (mTOR) is targeted in human and mouse NK cells (Figure 1A). The authors showed that treatment with TGF-β in vitro blocked interleukin-15 (IL-15)–induced activation of mTOR, resulting in downregulation of activatory receptors, reduced NK cell proliferation and cytotoxic activity [41]. Keskin et al. showed that TGF-β promotes conversion of CD16^+^ peripheral blood NK cells into CD16^-^ NK cells with similarities to dNK cells [43]. Again, TGF-β has been shown to upregulate CXCR3 and CXCR4 in tumor infiltrating NK cells [44], similar to what occurs for dNK cells [45,46,47].

Tumor-infiltrating and tumor-associated NK cells, in non-small cell lung cancer (NSCLC) patients, apart from functional anergy [21] can acquire the decidual-like CD56^bright^CD16^-^ phenotype, endowed with pro-angiogenic functions [2,20]. TGF-β has been identified as a master angiogenic-switcher in NKs, being able to polarize CD56^dim^CD16^+^cytolytic NK cells toward the CD56^bright^CD16^-^VEGF^high^PlGF^high^CXCL8^+^IFNγ^low^ NK cell subset [20] (Figure 1A).

Gao et al. demonstrated that TGF-β is able to convert NK cells (CD49a^-^CD49b^+^Eomes^+^) into ILC1 (CD49a^+^CD49b^-^Eomes^int^) [48]. Contrary to tumor immunization from NK cells, ILC1s were not able to control tumor growth. In this way, the tumor escapes the surveillance of the innate immune system by exploiting the sensitivity of NK cells to TGF-β to benefit from the plasticity of ILC1 in the TME [48]. Altogether, these data identify TGF-β as a relevant target to block the angiogenic switch in cancer patients and the inhibition of TGF-β signaling has been reported to preserve the function of highly activated, in vitro expanded NK cells in AML and CRC models [40] (Figure 1A).

Several cytokines, such as IL-2, IL-15, IL-21, IL-27, and IL-18, are generally referred as “immunostimulatory” for their ability of contrasting the effects of TGF-β1 [49]. Very unexpectedly, Casu et al. recently showed that IL-18, that is supposed to contrast immumosuppressive activities of TGF-β on NK cells, synergistically acts with TGF-β by contributing to the impairment of both NK cells recruitment and killing capability [49].

#### 2.1.2. HLA-G

HLA-G is an immunoregulatory class I MHC molecule that have been found to be expressed by decidual trophoblasts [50,51] and in diverse tumor tissues [51,52,53]. Tumor and serum HLA-G expression levels have been reported to be an independent marker of poor prognosis in several tumors, including NSCLC, ovarian, breast, colorectal, esophageal, gastric, hepatocellular and endometrial cancers [52,54,55,56,57]. Several controversies emerged around the expression of HLA-G by tumors [58]. In particular, some commonly used monoclonal antibodies to HLA-G give false positive results [58].

Immunosuppressive activities of HLA-G on NK cells act by interacting with immunoglobulin-like transcripts ILT-2, ILT-4 and killer Ig-like immunoglobulin receptor (KIR) 2DL4, whose interactions result in dampened NK cytotoxicity [59,60,61] (Figure 1B). Interaction between HLA-G with KIR2DL4 [62] has been documented to induce resting NK cell stimulation to produce several pro-inflammatory and pro-angiogenic factors, via induction of a senescence-associated secretory phenotype [45].

Therefore, in a model of metastatic ovarian cancer, HLA-G has been reported to support tumor progression by reducing NK cell cytotoxicity [63]. HLA-G have been also found to downregulate CCR2 and CXCR3, but not CXCR4 expression on CD56^bright^ NK cells, supporting the hypothesis that HLA-G is directly involved in NK cell recruitment by microenvironment [46] (Figure 1B).

#### 2.1.3. Prostaglandin E_2_

Prostaglandin E_2_ (PGE_2_) is associated with enhancement of cancer cell survival, growth, migration, invasion, angiogenesis, and immunosuppression [47] and largely produced within the TME [64,65]. PGE_2_ has been reported to inhibit NK cell cytotoxicity by reducing the expression of NKG2D, NCRs (NKp30, NKp44, and NKp46), and ADCC [66,67]. PGE_2_-induced NK cell anergy was associated a cAMP-mediated PKA type I-dependent mechanism following the binding of PGE_2_ on EP2 and EP4 receptors [68] (Figure 1D). Co-culture experiments also demonstrated relevant contribution of melanoma cancer-associated fibroblasts in mediating NK cell inhibition through PGE_2_ release [69] (see below). Given the recent finding in solid cancers, showing that anergic and IFNγ^low^ NK cells are subverted into pro-angiogenic NKs, it is conceivable that PGE_2_ can significantly contribute to the NK cell angiogenic switch. This will assume a dual role of PGE_2_ in supporting angiogenesis by direct action [70] and by polarizing anergic NK cells.

#### 2.1.4. Adenosine

Adenosine is a soluble immunomodulatory molecule acting through adenosine receptors (A1, A2A, A2B, and A3) that have been found to be expressed on multiple immune subsets including NK cells [71,72,73,74]. Adenosine peaks during decidualization [75] and up to 20-fold increases in the extracellular fluid of solid carcinomas has been reported [76]. Extracellular adenosine accumulation is partially sustained by hypoxia associated with the modulation of enzymes implicated in adenosine metabolism, like adenosine kinase and endo-5′ nucleosidase. Once released in the extracellular environment, adenosine has been reported to impair NK cell normal function by decreasing IL-2-dependent TNF-α secretion, inhibiting cytotoxic granule exocytosis, repressing perforin and Fas ligand-mediated cytotoxic activity [34] (Figure 1E). Many of these effects are attributed to stimulation of the cyclic AMP/protein kinase A pathway, following the binding of adenosine to A2A receptors on NK cells [34].

Adenosine has been reported to affect the expression of activating NK cell receptors by suppressing the release of cytotoxic cytokines in NK cells stimulated with IL-2/NKp46, an effect mediated by the combination of adenylyl cyclase and PKA I, through adenosine A2A receptor signaling [77] (Figure 1E). Immune suppressive activities of adenosine have been also described in NK cells stimulated with either IL-2, IL-15, or a combination of IL-12 and IL-15, by downregulation of the activating receptors NKG2D and NKp30 [78].

Very recently, Young et al. showed that A2AR adenosine signaling suppresses NK cell maturation in the TME [79]. The authors observed that engagement of A2AR acts as a checkpoint, by limiting NK cell maturation. They found that global and NK cell-specific conditional deletion of A2AR resulted in increased number of terminally mature NK cells at homeostasis, after reconstitution, and in the TME [79]. These results demonstrate that A2AR-mediated adenosine signaling acts as an intrinsic negative regulator of NK cell maturation [79].

#### 2.1.5. Extracellular Vesicles (EVs) and MicroRNAs (miRNAs)

Extracellular vesicles (EVs) are different submicron structures released by cells in a regulated fashion, that can be distinguished according to their size: apoptotic bodies (1000–5000 nm), microvesicles (200–1000 nm) and exosomes (30–150 nm) [80]. EVs are involved in intercellular communication between multiple cell types and substantially influence different physiological and pathological processes, including immune responses [81,82]. Diverse tumor cell-derived EVs (TEVs) can be differently up taken by NK cells [83]. EVs act as cargo containing miRNA that can bind to receptors of the Toll-like receptor (TLR) family on immune cells including NK cells, and are able to activate NF-κB and trigger a pro-metastatic inflammatory response [84]. TEVs can regulate NK cells, impairing their killing activity by down-regulating perforin/granzyme production and/or NKG2D ligand (NKG2DL) expression [44,85,86,87,88,89] (Figure 1C). The NKG2D/NKG2DL system plays an important role in tumor immune surveillance [87,88]. Berchem et al. showed that TEVs originating from hypoxic conditions exhibit strong immunosuppressive action on NK cells by delivering TGF-β, thus reducing NKG2D expression [90]. TEVs-associated TGF-β1 was linked with NK cell dysfunction in patients with AML [91]. Further studies reported that TEVs derived from diverse cancer cell lines, including mesothelioma, breast and prostate cancer cells, express NKG2DL and thereby down-regulate NKG2D expression on NK cells and CD8^+^ T cells, resulting in impaired cytotoxic effector functions [85,86]. It has also been shown that leukemia/lymphoma T and B cells secrete NKG2DL-expressing exosomes with the ability to impair the cytotoxic potency of NK and T cells from healthy donors [85,86]. In a similar manner, NKG2DL-bearing exosomes have been shown to be actively released by placental explants and play a role in the immune evasion of the fetus [92,93]. TEVs downregulate NKG2D expression on NKs also by shedding the NKG2DL on tumor cells, reducing NK cell activity [85,94,95]. TEVs may affect NK activity via other mechanisms including the down-modulation of IL-2-mediated pathways [96], suppressing perforin or cyclin D3 production [91] and Janus kinase (Jak)3 activation resulting in a failure of NK cell-mediated cytolysis [91].

Recent studies reported that EVs can activate immune cells. Viaud et al. demonstrated that dendritic cell (DC)-derived exosomes promote an IL-15Rα and NKG2D-dependent proliferation and activation of NK cells in a murine in vivo model, resulting in tumor regression. They also showed that a DC-derived exosome-based vaccine restored NKG2D-dependent functions of NK cells in half of the tested melanoma patients [63]. Oral cancer-derived EVs were able to promote the biological functions of NK cells, including proliferation, release of perforin and granzyme M, enhancing the cytotoxicity, thereby promoting their functions [97]. In another study, it was shown that EVs derived from genetically modified cells expressing IL-15, IL-18, and 4-1BBL, similar to their host cells, were able to increase NK cell cytotoxicity in tumor cells following a short time treatment (4 h). However, with an extended treatment time (48 h), these EVs inhibited the cytotoxicity of NK cells by acting on NKG2D, pointing out the dual effects of EVs on NK cells [98].

A better understanding of the mechanisms by which EVs influence the NK cell phenotype and function can open new possibilities for the use of EVs in controlling immune responses, either as a targeted therapy or as an adjuvant to modulate immune-based anti-cancer treatments.

MicroRNAs (miRNAs) are conserved non-coding single small in length (~22 nucleotides) stranded RNA molecules, now recognized as important regulators of several cellular processes, including immune function and cancer survival [99,100,101]. Beside their role in cancer, miRNAs have been shown to play a critical role in NK cell activation, effector response, and dysregulation in malignancies [102,103] (Table 1). However, the mechanism by which miRNAs regulates NK cell function is largely unknown. MiRNA profile impact on NK cell development and function was established in several studies through disrupting global miRNAs in mouse NK cells, resulting in decreased NK cell survival, maturation, and proliferation [103,104]. Pesce et al. found differentially expressed miRNAs in human NK cell subsets providing valuable clues of miRNA regulation in human NK cell maturation [105]. Specific miRNAs linked to the development and prognosis of several malignancies such as miR-15/16, miR-24, miR-29b miR-155, miR-150, miR-181, miR-483-1, miR-583 directly mediate NK cell development and differentiation by targeting specific genes [106,107] as described in Table 1. For instance, miR-181a/b regulates NK cell differentiation by targeting Notch signaling and upregulate IFN-γ production [108], while miR-15/16 family has been shown to directly target the murine IFN-γ 3′ UTR and the transcription factor c-Myb (Myb) affecting NK cell maturation program [106,109]. Additional miRNAs such as miR-146a negatively regulate IFN-γ production in NK cells by targeting IRAK1 and TRAF6, with subsequent inhibition of the NF-κB signaling cascade [110]. Similarly, it was shown that miRNA-146a reduce NK cell-mediated cytotoxicity and the production of interferon IFN-γ and TNF-α by targeting STAT1 [111]. In human NK cells, miRNA-155 overexpression was correlated with enhanced IFN-γ production and to directly downregulate SHIP1 (SH2-containing Inositol 5′-Phosphatase 1), a growth receptor expressed in CD56^bright^ and CD56^dim^ NK cell subsets [112,113,114]. In an AML mouse model, NK cells displayed high levels of miRNA-29b and reduced T-bet and EOMES associated with a block of CD56^bright^ NK cell and cytokine-secreting potential [115]. A knockout mouse model of miRNA-29b and forced overexpression of T-bet and EOMES, primed NK cell development suggesting a key role of miRNA-29b in tumor evasion from NK cell surveillance [115].

De-regulation of miRNAs that interfere with NK cell cytolytic activity (by targeting directly or indirectly granzyme B and perforin), such as miR-27a-5p, miR-146a-5p, miRNA-150, miRNA-519a-3p, and miRNA-615-5p, has been observed in different tumors [105,116,117,118,119]. Interestingly, Regis et al. showed that miR-27a-5p, apart from impacting on the effectors function of NKs, downregulates the expression of CX3CR1 in primary human NK cells via TGF-β. Affection of CX3CL1/CX3CR1, strongly impact on the recruitment of CD56^dim^ NK cells, as a tumor strategy to escape immune recognition [120].

In NK cells of hepatocellular carcinoma (HCC) patients it was suggested that miRNA-182 may augment NK-cell cytotoxicity against liver cancer through perforin-1 up-regulation and by modulating NKG2D and NKG2A expressions [121].

The ability of miRNAs to hit simultaneously multiple tumor-promoting pathways make them attractive targets and miRNA regulation of NK cells may be a new therapeutic approach to treat cancer.

### 2.2. Cell-to-Cell Interactions

#### 2.2.1. Macrophages

Among tumor-infiltrating innate immune cells, macrophages represent the best characterized and they are directly involved in many processes contributing to tumor initiation and progression [127]. Tumor-associated macrophages (TAM) with features of M2 macrophages can promote immunosuppression, fostering tumor invasion, angiogenesis, and lymphangiogenesis; thus, they support tumor progression and metastasis and are generally associated with poor prognosis [127,128,129]. Monocytes and macrophages are recruited in the tumor stroma to become TAMs by several inflammatory mediators, such as chemokines: CCL2, CCL5, and CXCL12, produced in TME by several cellular components including NK cells. Macrophages can inhibit NK cell-mediated activity by the release of soluble factors and cell-to-cell contact such as the expression of checkpoint blockade PD-L1 [130] (Figure 2A); on the other hand, NK cells could produce diverse inflammatory factors able to commit macrophage function. In this context, Bellora et al. studied in vitro cellular interactions between macrophages and NK cells [131]. These authors showed that following LPS stimulation both M0 and M2 macrophages became capable to fully activate NK cell by a cell-to-cell contact-dependent mechanism and through the DNAM-1 and 2B4 surface receptors [132] (Figure 2A). Further results by Mattiola et al. corroborated and deepened these findings [133].

Macrophage-NK cell cross-talk in tumor setting has been investigated in ascites of ovarian cancer patients [134]. Results have revealed that whereas untreated TAMs induced inhibition of NK cell activity, the LPS stimulation of TAMs was able to restore NK cell effector functions against the NK-resistant OVACR-3 tumor cell line [134]. TAMs can dampen NK cell anti-tumor functions by realizing TGF-β and PGE_2_. Recently, it has been shown that in early lung adenocarcinoma lesions, a low level of activation of intralesional NK cells correlated with inactivated T cells, high grade of Treg cells and macrophages expressing the PPARγ^high^ suppressive phenotype [135]. Finally, new data have been added on the complex interplay between macrophages and NK cells, showing the relevance of the IL-1R8 on NK cell function during interaction with both macrophages and DCs. Indeed, the authors showed that IL-1R8-deficient NK cells in an IL-18-dependent manner were responsible of the anti-metastatic effect in two distinct tumor murine models [136] (Figure 2A). DCs can also play a key role in the regulation of NK cell activities in anti-tumor responses, mainly by a positive effect [137,138,139,140,141].

#### 2.2.2. Neutrophils

Neutrophils are implicated in the acute phase of inflammation and they can acquire the capacity to extend their survival and promote cellular interaction with other leucocytes [142,143]. In the contest of the TME, neutrophils undergo a molecular reprogramming in response mainly to TGF-β that leads to the establishment of tumor-associated neutrophils (TANs or N2), endowed with pro-tumor features and pro-angiogenic capabilities [3,5,144,145,146,147]. The interplay between neutrophils and NK cells is becoming increasingly appreciated [148]. In different contexts the relationship between neutrophils and NK cells could result in both activation and suppression [149]. Several reports have documented that ROS and arginase I from neutrophils could induce functional impairment in NK cells [150,151] (Figure 2B). Interestingly, it has been reported that CD56^bright^CD16^-^ NK cell subset is resistant to neutrophil-derived ROS, probably due to high anti-oxidative capacity of this subset than CD16^+^ NK cells, thereby TANs can induce selection and expansion of CD56^bright^CD16^-^ NK cells [152]. Moreover, TANs can secrete CCL2 and CCL5 promoting NK cell recruitment in the TME [149,153,154] (Figure 2B). At the same time, CD56^bright^CD16^-^ NK can release CXCL8 and favor neutrophil accumulation. It has been recently shown that NK cell can regulate tumor-promoting inflammation through functional modification of neutrophils. In a sarcoma transplantable murine model, NK cells have been shown to regulate neutrophil activities via IFN-γ, likely by an indirect mechanism and through the IL-17 axis. Therefore, tumor progression in a NK cell-depleted host is blocked when the IL-17A neutrophils axis associated with high levels of neutrophil-dependent VEGF are absent [155]. Finally, Spiegel et al. showed that TANs can support metastasis by interacting with NK by inhibiting NK cell-mediated cytotoxic activities and thus protecting intraluminal trapped tumor cells [156].

#### 2.2.3. Mast Cells

Mast cells (MCs) are granulated tissue-resident cells, originating from hematopoietic innate immune system, which can exert protective immune responses against viral and microbial pathogens [157,158], but also adverse reactions in allergic diseases. In cancer patients, increased numbers of MCs have been detected inside the tumor and in peritumor tissues [159]; this increment is more evident in angiogenesis-associated with vascular tumors, such as hemangioma and hemangioblastoma, as well as with several hematological and solid tumors. The increased amount of MC in these tumors was associated with raised neovascularization, presence of MC-derived VEGF and FGF-2, and poor prognosis [160]. On the other hand, anti-tumor functions of MCs have been identified deriving from their ability to trigger target cell cytotoxicity by releasing TNF-α, by production of ROS, or by NK cells recruitment, as shown in a MC-derived CCL3-dependent orthotopic melanoma model [161]. MCs can induce NK cell recruitment also by the release of CXCL8 [162] (Figure 2C). The TGF-β production by MCs in the TME could be responsible of the switch from CD56^dim^CD16^+^ NK cell subset to the noncytotoxic CD56^bright^CD16^-^ cell subset (Figure 2C). MCs can release adenosine in the extracellular milieu that has been reported to dampen NK cell activity by downregulating NKG2D and NKp30 expression and TNF-α/IFN-γ release [77,78] (Figure 2C).

Recently, the relevance of potential cross-talk between MCs and NK cells in the TME has been reported and in particular in the tumor angiogenic process [160,163]. Moreover, it has been shown that LPS-stimulated bone marrow MCs trigger cell contact-dependent IFN-γ secretion by NK cells, and this activation process is partly mediated by OX40L expression on MCs [164]. Several lines of evidence point out the role of NK cells and MCs in tumor growth and angiogenesis, given their ability to synergize in different pathological conditions [163].

#### 2.2.4. Cancer-Associated Fibroblasts

Cancer-associated fibroblasts (CAFs) are the most abundant cell type within the active stroma of many cancer types [165,166,167]. Pro-tumorigenic activities of CAFs have been demonstrated and include induction of cell proliferation and progression by favoring metastasis [165,166,167]. CAFs are able to orchestrate tumor progression via secretion of various growth factors, cytokines, chemokines, and the degradation of extracellular matrix. Through gene expression and mass spectrometry analyses, several studies identified immunomodulatory activities of CAFs showing that their secretome is rich in several cytokines and chemokines endowed with immunosuppressive actions (IL-6, TGF-β, IL-1β, IL-10, IDO, and PGE_2_), inflammatory cell recruitment (CXCL1, 2, 5, 6, 9, 10, 12, CCL2, 3, 5,7, 20, and 26), and pro-angiogenic activities (VEGF, CXCL8, and FGFs) [168,169,170,171]. Indeed, CAFs exert relevant pro-tumorigenic activities supporting the cell metabolic reprogramming, as a consequence of the Warburg effects [172,173,174].

CAF/NK cell cross-talk results in the induction of NK cell immunosuppression by downregulation of NKG2D expression and functional anergy. CAF-derived TGF-β can acts on NK cells by inhibiting NK cell cytotoxicity, block IFN-γ release and can convert cytotoxic CD56^dim^CD16^+^ NKs towards the pro-angiogenic CD56^bright^CD16^-^VEGF^high^PlGF^high^CXCL8^+^IFN-γ^low^ NKs [2,20] (Figure 3A). In melanoma patients MMPs released by CAFs support NK cell immunosuppression by selectively cleaving MICA and MICB on tumor cells [175]. CAFs are also an abundant source of PGE_2,_ that has been reported to switch off NK cell activities in several cancers, by downregulating the expression of NKG2D, NKp30, NKp44 and decreasing perforin/granzyme B release [69,176,177] (Figure 3A).

#### 2.2.5. Endothelial Cells

Interactions between NK and endothelial cells occur as early events during the immune surveillance of tissues, inflammatory responses and wound healing. In this scenario, most studies have been addressed to dNKs [16,17]. dNK cells have a CD56^superbright^CD16^-^KIR^+^ phenotype, are poorly cytotoxic and produce large amounts of pro-angiogenic factors, such as VEGF, PlGF, CXCL8, and IL-10 [16,17]. Early on pregnancy, accumulating dNK represent 70% of the local lymphocytes and 30–40% of all decidual cells. We were the first in describing the pro-angiogenic activities on NK cells in different cancer types [2,5,20,22,29,178].

In patients with NSCLC we identify a polarized NK cell subset, defined as CD56^bright^CD16^-^VEGF^high^PlGF^high^CXCL8^+^IFN-γ^−^ (Figure 3B) able to induce human umbilical vein endothelial cell (HUVEC) migration and the formation of capillary like structures [20]. These pro-angiogenic features were observed in NK cells from NSCLC tissues and patients’ peripheral blood, suggesting that angiogenic switch in NSCLC NKs occurs already at systemic level [2,20]. Interestingly, we found that NKs from patients with squamous cell carcinomas exhibited even higher pro-angiogenic activities as compared with those with adenocarcinomas [20].

NK cells from malignant pleural effusion have shown to be endowed with a decidual-like pro-angiogenic polarization, by acquiring a CD56^bright^CD16^-^CD49a^+^VEGF^+^ phenotype and are able to support the in vitro capillary-like structure formation in HUVECs [178]. Induction of a pro-angiogenic and decidual-like NK cell phenotype has been observed also in CRC patients. TINKs in CRC patients are polarized towards the CD56^bright^CD16^-^CD9^+^CD49a^+^ subset (Figure 3B) and can induce in vitro HUVEC proliferation, migration, and vessel formation [22]. Moreover, CRC TANKs, apart from exhibiting a decidual-like CD56^+^CD9^+^CD49a^-^ phenotype, can release large amount of pro-angiogenic factors including VEGF, Angiogenin, Angiopoietin-1, CXCL8, MMP1, MMP9, TIMP-1, and support angiogenesis in vitro [22]. Interestingly, this CRC TANK polarization was associated with increased level of STAT3 and STAT5 and of note, inhibition of this axis by the anti-psychotic agent Pimozide, resulted in blocked release of VEGF and Angiogenin and functional inhibition of their pro-angiogenic activities [22].

## 3. NK Cells as a Therapeutic Tool: Current Strategies

The therapeutic potential of NK cells in cancer immunotherapy was firstly highlighted by Miller et al. in 2005, showing that the infusion of short-term IL-2-activated allogeneic haploidentical NK cells induce the remission in patients with refractory leukemia [179].

Despite this evidence, several concerns about the real feasibility of using NK cells in cancer immunotherapy have been pointed out. Indeed, the most of therapies are based on the antigen specificity that so far has been considered a unique property of T and B cells, even if the existence of a peculiar subset of Ly49^+^ NK cells characterized by a T cell-like immune memory has been recently demonstrated [180,181]. Moreover, the limited clonal abilities in vivo and tumor-homing capacity of NKs, along with their substantial phenotypic differences that cannot be recapitulated by animal models [182], still represented other relevant issues to study NK cell-based immunotherapeutic approaches.

Nevertheless, new advances in NK cells manipulation and novel findings of molecular mechanisms involved in NK cell anti-tumor activity have allowed the development of several approaches aimed at enhancing NK cytotoxicity against cancer cells.

### 3.1. Cytokines That Boost NK Cell Anti-Tumor Immunity

Due to its ability to enhance T cell as far as NK cell proliferation, homeostasis and cytotoxicity, IL-2 was the first cytokine employed in the clinic to boost immune responses in cancer patients (Figure 1A). Despite the high expectations, results from these studies demonstrated that the therapeutic anti-tumor potential of IL-2 administration was limited and especially when used at high doses, a relevant toxicity was observed [182]. Thus, other studies have focused on the use of low doses of IL-2 or antibody–cytokine fusion proteins also designated as immunocytokines characterized by a lower toxicity profile [183]. However, results obtained from these studies demonstrated that the enhancement of NK cell function was associated with Treg cell mobilization. In particular, Hirakawa et al. showed that low-dose of IL-2 induce STAT5 activation in Helios^+^ Treg cell and CD56^bright^CD16^–^ NK cells in vitro and the selective expansion of these cell subsets was observed in GVHD patients upon IL-2 treatment [184]. Moreover, they also found the upregulation of Ki67 and CTLA-4 in NK cells and the increased expression of PD-1 in CD4^+^ and CD8^+^ T cells [184] (Figure 4A). Recently, a CEA-targeted IL-2 variant-based immunocytokine that abolishes CD25 binding, has been used to overcome the concomitant IL-2-based suppressive Treg cell activation with encouraging results [185]. Recently, a novel IL-2 variant, termed super-2, with increased binding affinity for IL-2Rβ has been proposed by Levin et al. who demonstrated in vivo the ability of their modified IL-2 to induce a higher expansion of cytotoxic T cells and a decreased activation of Treg cells [186]. To overcome Treg cell activation, IL-15, a cytokine that stimulates CD8^+^ T cells and non-terminally differentiated NK cells has been proposed as immunotherapeutic agent. The single-chain recombinant IL-15 (scIL-15) as far as soluble IL-15Rα have been used to stimulate NK proliferation and functions both in solid and hematologic malignancies [187,188] and are currently under evaluation by several ongoing clinical trials. An alternative cytokine-based approach used to boost NK cell cytolytic activity is IL-12 administration. IL-12 has been shown to promote IFN-γ release, migration and NK-mediated ADCC (Figure 4A), due to the induction of specific adhesion molecule including the selectin CD62L and the upregulation of KIRs and CD16 [189], however additional studies are needed to clarify the value of IL-12 use in the context of cancer therapy [190,191,192].

### 3.2. Drugs Enhancing NK Cell Anti-Tumor Activity

Pre-clinical studies using thalidomide derivatives demonstrated that lenalidomide and pomalidomide can indirectly enhance NK cell cytotoxicity by activating the intracellular signaling of phosphoinositide-3 kinase (PI3K), followed by nuclear translocation of nuclear factor of activated T cells 2 (NFAT2) and activator protein 1 (AP-1) allowing the release of IL-2 and IFN-γ from T cells and DCs [193,194] (Figure 4B). Moreover, lenalidomide has been shown to induce the degradation of Ikaros and Aiolos, two transcription factors repressing IL-2 production in T cells [195] and concomitantly to up-regulate the expression of the NKG2D, DNAM-1 activating receptor, and ligands MICA and PVR/CD155 in human multiple myeloma (MM) cells [196].

On the basis of these pre-clinical and in vitro evidence, some clinical trials are focusing on the role of lenalidomide in enhancing NK cytotoxic abilities. However, a very recent study evaluating the effects of lenalidomide administration in MM patients showed that lenalidomide treatment neither activated NKs nor enhance degranulation or IFN-γ release by NK cells [197]. Another approach to increase the cytotoxicity of NKs is the disruption of inhibitory KIRs through mAb-mediated blockade. In this context, different mAbs have been developed, termed 1-7F9, IPH2101, and IPH2102 (Figure 4B). Two phase I clinical trial have demonstrated the low toxic profile of KIR-specific mAbs in cancer patients, while no significant changes were observed in the number of NK cells or have shown any reduction in KIR2D-positive NK cells upon treatment. Only a transient increases of serum TNF-α and MIP-1β was found in treated patients and a transient induction of CD69 expression on NK cells was observed [198,199].

### 3.3. Immune-Checkpoints Inhibitors

Benson et al. demonstrated that using the novel anti-PD-1 antibody, CT-011, is able to enhance NK cell functions against autologous and primary MM cells and increase NK trafficking by up-regulating CXCR4 [200] (Figure 4C). It has been demonstrated that disrupting PD-1 inhibitory pathway improved IFN-γ release by NK cells without enhancing their cytotoxic abilities [201] Figure 4C). Moreover, the ex vivo use of anti–PD-L1/PD-L2 mAbs was able to partially restore the degranulation of PD-1^+^ NK in presence of PD-L1/PD-L2^+^ OVCAR5 target cells [130]. Zhang et al. have also recently demonstrated that the blocking of TIGIT can prevent NK cell exhaustion and trigger NK cell-dependent tumor immunity in several mouse models [202].

### 3.4. Bi- and Tri-Specific Killer Engagers

These small molecules consist of a single heavy (VH) and light chain (VL) of the variable region of CD16 linked to one (BiKE) or two (TriKE) variable portions of several tumor antigens (Figure 4D), including CD20 and CD19 for non-Hodgkin’s lymphomas [203,204,205,206], CD19 and CD33 for different types of leukemia [207], CD30 for Hodgkin’s disease [208], EGF-R for EGF-R^+^ tumors [209], HER2/neu for metastatic breast cancer [210,211] and EpCAM for prostate, breast, colon, head, and neck carcinomas and MOV19 on ovarian cancer [212]. These agents improve cytotoxicity of NK cells by inducing the strong release of cytokines by NK cells and enhancing the NK-mediated ADCC [213].

### 3.5. Drugs Sensitizing Tumors to NK Cells

Drugs able to increase the susceptibility of tumor cells to NK cytotoxic effects have been recently proposed as an alternative approach. In particular, proteasome inhibitors, such as bortezomib, have been demonstrated to be able to induce the expression of tumor necrosis factor–related apoptosis-inducing ligand (TRAIL) receptors on tumor cells promoting their lysis by NK cell [214]. Moreover, in vitro treatment with bortezomib was also able to increase the expression of MICA/B, Nectin-2, and PVR expression on MM cells enhancing the sensitivity of MM cells to NK cell-mediated lysis [215] (Figure 4E).

Accordingly, in vitro studies using histone deacetylase (HDAC) inhibitors, including valproic acid, have demonstrated its ability to upregulates MICA/B and UL16-binding protein (ULBP) 2 on MM cells by inducing the phosphorylation of ERK ½ and in turn promoting their lysis by NKs [216] (Figure 4E). Skov and colleagues also demonstrated that the treatment of tumor cells with PXD101, suberoylanilide hydroxamic acid (SAHA), and trichostatin A, three HDAC inhibitors, induce MICA/B expression on Jurkat T cells by activating GSK-3 kinase signaling [217] and becoming targets for NKG2D-expressing cells like NK cells (Figure 4E).

### 3.6. Adoptively Infused NK Cells

Allogeneic hematopoietic cell transplantation (HCT) is one of the most efficient therapeutic options in the context of myelodysplastic syndrome (MDS), AML, or chronic myelogenous leukemia (CML). This approach can result in durable remission of malignancies due to the development of graft versus leukemia (GVL) even if relapse is described in 40% of patients [218]. Immunotherapeutic strategies using adoptively transferred NK cells are particularly relevant for the possibility to pre-activate and manipulate NK cells prior to infusion (Figure 5A). Adoptive transfer of short-term allogeneic NK cells stimulated with 1000 IU/mL IL-2 for 8–16 h prior to infusion has been demonstrated to induce a clinical response in AML and MM patients [179,219]. The NKAML trial (Pilot Study of Haploidentical NK Transplantation for AML) demonstrated that the KIR-HLA-mismatched donor NK cell infusion reduces the risk of relapse in childhood AML with limited non-hematologic toxicity [220]. Shaffer et al. in a phase 2 study demonstrated the safety of haploidentical NK cell infusion after allogeneic HCT in 8 patients with relapsed or progressive AML or MDS. In this context, the authors showed transient responses in two patients and morphologic resolution of dysplasia in a third one [218]. Moreover, it has also been demonstrated that an improvement of the clinical response occurs whether adoptive cell transfer was followed by IL-2, which promote in vivo expansion of infused NK cells [179,221]. In contrast to short term activation, ex vivo expansion of peripheral NKs with media containing cytokines such as IL-2 and IL-15 allows to obtain high number of activated NK cells [222] (Figure 5A).

### 3.7. Genetic Modification of NK Cells

Some studies have been focused on cytokine gene transfer to promote NK cell survival (e.g., IL-2, IL-12, IL-15, and stem cell factor). In particular, engineered NK cell lines displayed an increased cytotoxic ability as well as proliferative rate, survival and in vivo anti-tumor activity [222]. In other investigations, NK cells were genetically manipulated to express a high-affinity variant of CD16a and it was demonstrated that a single-nucleotide polymorphism in CD16 that results in an amino acid substitution at position 158 (CD16-F158V) can bolster ADCC in vivo [223] (Figure 5B).

### 3.8. Chimeric Antigen Receptor (CAR)-Engineered NK Cells

Several pre-clinical and clinical trials have focused on the development of Chimeric Antigen Receptor (CAR)-modified T cells, while little is known on CAR-engineered NK cells. Studies focused on the therapeutic application of CAR-engineered NK cells use peripheral blood (PB-NK cells) or umbilical cord blood (UCB-NK cells) as source of NK cells (Figure 5C). This strategy showed a limited efficacy against solid tumors. Moreover, to collect sufficient number of NK cells, donors have to undergo repeated leukaphereses and have to be expanded in vitro by using feeder layers. In addition, NK cells collected are extremely variable, expressing different levels of markers such CD16 and KIRs [224]. As an alternative, the NK-92 cell line was also employed in clinical trials. The main concern on this strategy is represented by their mild stimulatory anti-cancer abilities, even if they still represent a valid therapeutic tool [224]. It has been demonstrated that also induced pluripotent stem cells (iPSC) can be used to generate a homogeneous NK cell population that can be genetically modified using both viral and non-viral methods to obtain CAR-NKs [225]. Moreover, Li et al. demonstrated that NK cells derived from human iPSCs expressing the transmembrane domain of NKG2D, the 2B4 co-stimulatory domain, and the CD3z signalling domain display a higher anti-tumor ability compared with PB-NK cells and T-CAR-iPSC-NK cells in an ovarian cancer xenograft model [225]. Most of the studies using CAR-engineered NKs have addressed their attention on CAR against CD19 and CD20 targeting B cell malignancies [226], demonstrating that CAR-expressing NK-92 cells effectively kill chronic lymphocytic leukaemia (CLL) cells and that such a cytotoxic response is significantly higher than that resulting from ADCC mediated by mAbs [226]. Esser et al. generated NK-92 cell expressing a disialogangliosideGD(2)-specific CAR for neuroblastoma (NB) treatment. These authors have demonstrated that CAR expression by gene-modified NK cells promote the recognition and elimination of NB cells. The efficacy of the treatment with GD(2)-specific NK was strictly dependent by antigen recognition and it can be reverted using anti-GD(2)or anti-idiotypic antibodies [227]. Recently, CXCR4-engineered NK cells with concomitant expression of EGFRvIII-specific CAR were generated by Muller et al. aiming to kill glioblastoma cells. By using in vitro and in vivo approaches, they have demonstrated that the expression of CXCR4 improve the migration to U87-MG glioblastoma cells resulting in a stronger anti-tumor effect compared with control groups [228]. Schönfeld et al. showed that NK-92/5.28.z ErbB2 (HER2)-specific NK cells efficiently lysed ErbB2-expressing tumor cells in vitro and were also able to reduce in vivo lung metastasis in a renal cell carcinoma model [229].

## 4. Conclusions

Altogether, it is clear that NK cells, besides the well-established anti-tumor effect, may support cancer both by immunosuppression and by supporting tumor angiogenesis. This unfavorable dark side of NK cells tightly depends on the interactions with the various cellular components of the host, placing TME as the major player in the tumor-immune escape and immune cell pro-tumor/pro-angiogenic-features. The cell-to-cell contact, cytokines, chemokines, immunomodulatory molecules, extracellular vesicles, play a key role in determining the final NK cell-mediated effect, leading to either proliferation or elimination of tumor cells. Thus, we propose NK cells as a relevant host-dependent hallmark of cancer and a key cellular paradigm in tumor progression and angiogenesis; thus, NK cells should be considered as a suitable target to modulate the immunosuppressive TME and try to trigger a more potent anti-tumor response.

## Figures and Tables

**Figure 1 cancers-11-00461-f001:**
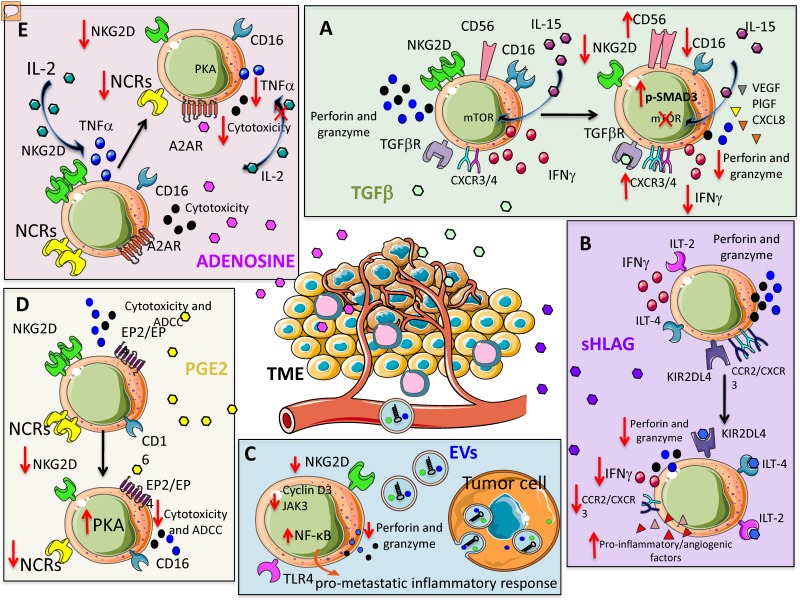
Soluble factors within the tumor microenvironment (TME) orchestrating natural killer (NK) cell pro-tumor features. Several soluble or microvesicle/exosome-associated factors can impair NK cell-mediated anti-tumor activities. These factors can be produced by different components of the TME besides tumor cells. (**A**) TGF-β can inhibit interleukin-15 (IL-15) triggering of NK cells, impairing the mTOR signaling; this results in the reduction of NKG2D expression and consequent killing of NKG2DL^+^ tumor cells; also, CD16, perforins, granzymes and IFN-γ are downregulated. This revert NK cells to a pro-angiogenic phenotype characterized by the secretion of VEGF. (**B**) Soluble HLA-G interacting with the KIR2DL4, ILT-4 and ILT-2 inhibitory NK cell receptors shape the behavior of NK cells from cytotoxic to pro-angiogenic. (**C**) Tumor-derived extracellular vesicles (EVs) and exosomes expressing NKG2DL can induce NK cell anergy because they interact with NKG2D on NK cell surface; this impairs the binding of NK cells with NKG2DL present on tumor cells. (**D**) PGE_2_ recognizes EP2/EP specific receptors on cytotoxic NK cells leading to the decrement of expression of several activating receptors such as NKG2D and NCR. This leads to the inhibition of tumor target recognition and killing. (**E**) Likewise, PGE_2_, adenosine upon binding with A2AR receptors on NK cells induces the downregulation of NKG2D and NCR; this results in a reduced killing of tumor target cells expressing NKG2DL and/or NCRL.

**Figure 2 cancers-11-00461-f002:**
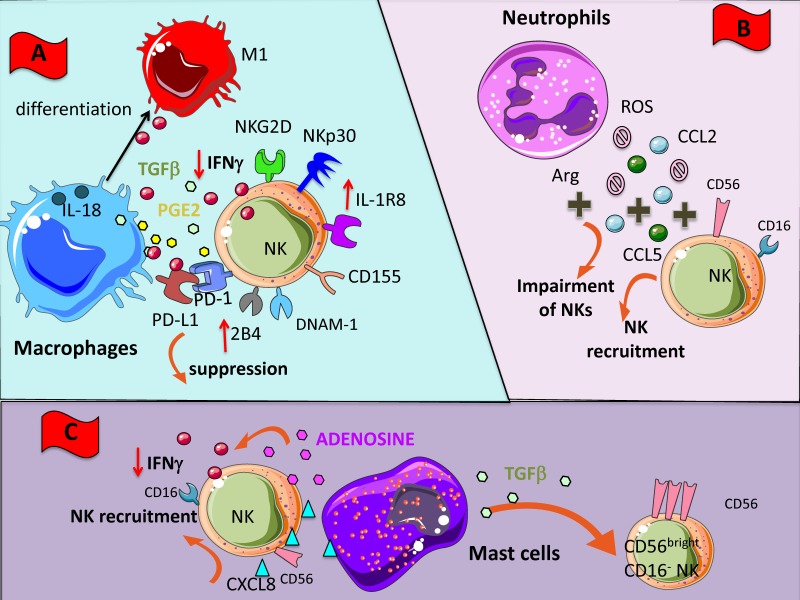
NK cross-talk with innate immunity cellular components of TME. Within TME, NK cells can encounter different cells such as macrophages (**A**), neutrophils (**B**), and mast cells (**C**). All these cells have been shaped by the TME to exert an inhibitory effect on NK cells through the direct cell-to cell interaction or the soluble factors listed in Figure 1. (**A**) M2-like macrophages can deliver negative signals to NK cells by the release of TGF-β and PGE_2_ but also by the binding of PD-L1 to PD1; these interactions lead to downregulation of several NK cell activating receptors, such as NKG2D, NCR and DNAM1. (**B**) Neutrophils can impair NK cell functions via reactive oxigen species (ROS) and arginase (Arg) activity; they also secrete CCL2 and CCL5 chemokine favoring NK cell tissue localization; in turn, NK cells release CXCL8 which attract to the tumor site other neutrophils amplifying the inhibition. (**C**) Mast cells produce TGF-β and respond to CXCL8 produced by NK cells; this TGF-β inhibits the cytolytic activity of NK cells by the downregulation of activating receptors, similar to the effect of macrophages.

**Figure 3 cancers-11-00461-f003:**
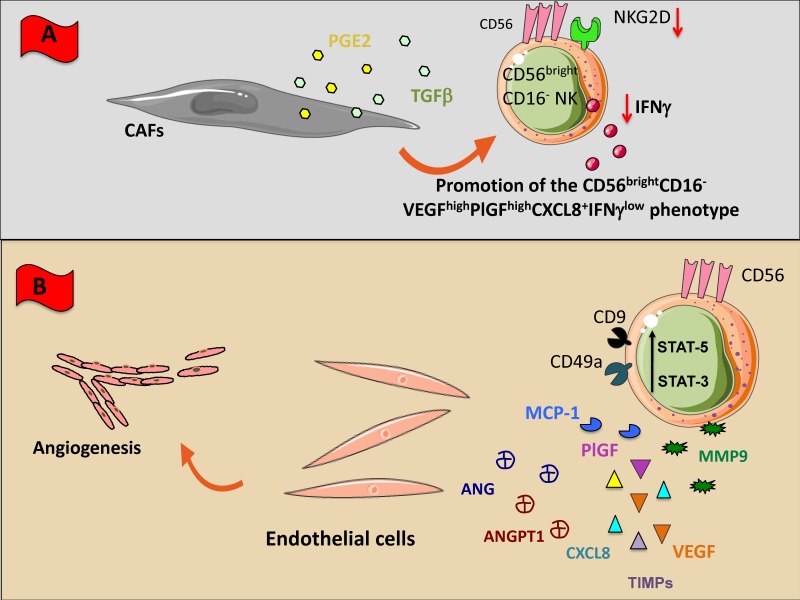
NK cross-talk with stromal (non-immune) cellular components of TME. Within TME, NK cells can encounter different non-immunological stromal cells such as macrophages (**A**), cancer associated fibroblasts (CAFs), and endothelial cells (**B**). All these cells have been shaped by the TME to exert an inhibitory effect on NK cells through the direct cell-to cell interaction or the soluble factors listed in Figure 1. (**A**) CAFs downregulate NK cell functions releasing inhibiting factors including TGF-β, and PGE_2_; (**B**) Tumor infiltrating NK cells and NK cells present in peripheral blood express high levels of CD56 but negative or low expressing CD16 and cytolytic behavior; these NK cells typical of tumor patients produce several factors such as VEGF, Angiogenin (ANG), Angiopoietin-1 (ANGPT1), PIGF, CXCL8, and metalloproteinase which stimulate endothelial cell growth and angiogenesis.

**Figure 4 cancers-11-00461-f004:**
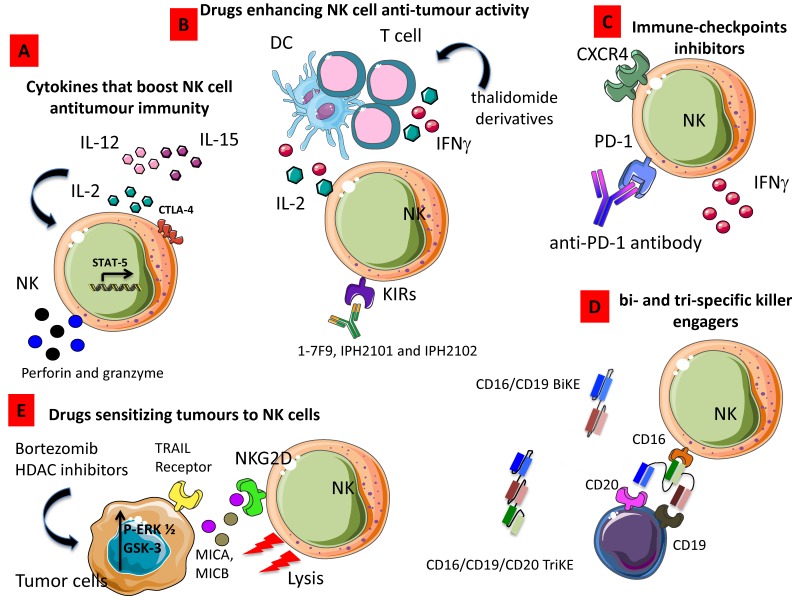
Strategies enhancing NK cell anti-tumor activity. (**A**) Cytokines such as IL-2, IL-15, and IL-12 alone or in combination can increase production and release of cytolytic granule content boosting NK cell anti-tumor immunity. (**B**) Drugs, including thalidomide derivatives, enhance the production of IFN-γ, thus triggering NK cell-mediated cytolysis. (**C**) Antibody directed to immune-checkpoints inhibitors like PD-1 can relieve the brake to NK cell cytolysis. (**D**) Bi- and tri-specific killer engagers strongly activate NK cell-mediated killing of tumor target cells. (**E**) Drugs able to sensitize tumors to upregulate ligands of activating receptors can increment killing of tumor cells; indeed, histone deacetylase inhibitors (HDAC) trigger the expression of NKG2DL such as MICA and MICB on tumor cells, these cells are more easily recognized and killed by NKG2D^+^ NK cells.

**Figure 5 cancers-11-00461-f005:**
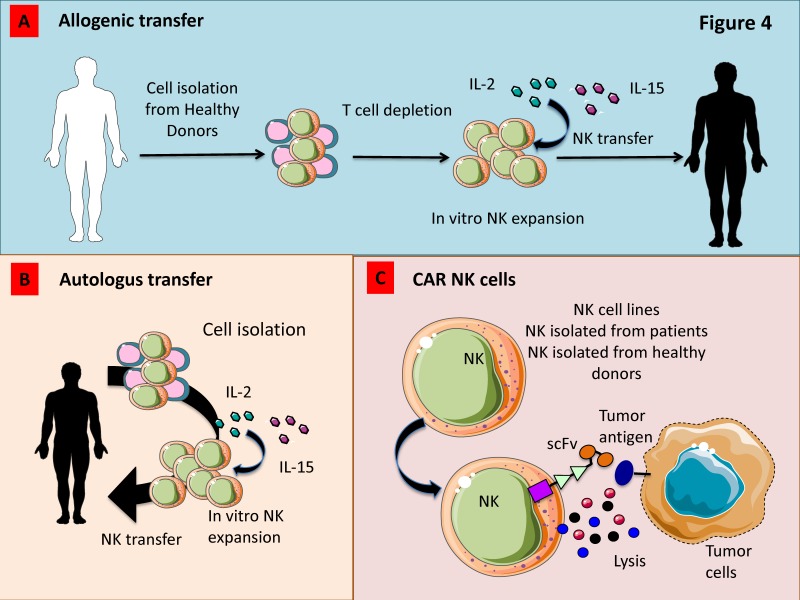
Strategies to increase the anti-tumor activity of adoptively transferred NK cells for anti-cancer therapy. (**A**) Allogenic NK cells expanded ex vivo with IL-2 and/or IL-15 show a strong anti-tumor effect in patients which do not express the HLA-I allele recognized by inhibitory HLA-I receptors present on donor NK cells. (**B**) Also, autologous NK cells expanded ex vivo can be used to kill tumor cells. (**C**) Novel tools such as Chimeric Antigen Receptors (CAR) transduced into NK cells isolated from either healthy donors or patients can trigger transferred-NK cells to kill tumor cells. To avoid the variability of the NK cell activity from donor to donor, some NK cell lines have been transduced with CAR and used in clinical trials.

**Table 1 cancers-11-00461-t001:** Effects of microRNAs (miRNAs) on NK cell anti-tumor activities.

miRNA and Role in NK Cells				
miRNA	Target	Role in NK Cell	System	References
miR-15/miR-16	c-Myb IFN-γ	NK cells maturation; IFN-γ production	mouse	[106,109]
miR-24	Paxillin	Inhibition of IFN-γ, TNF-α and decreased cytotoxicity	human	[107]
miR-27a-5p	Prf1 GzmB Responsiveness to CCL4 (MIP1β) and CXCL8 (IL-8)	NK cell cytoxicity	human	[117]
miR-29b	TBET EOMES	Terminal differentiation; reduce CD56^bright^ NK cell subset	human mouse	[115,122]
miR-146a	IRAK1 TRAF6 STAT1	IFN-γ and TNF-α production	human	[110,111]
miR-146a-5p	KIR2DL1 KIR2DL2	NK cell activation, KIR and perforin expression	human	[105]
miR-150	c-Myb.	Activation and maturation; increased expression of GZMB, KIR2DL2, CD16, CD56, NKG2D, NKp46	human mouse	[123,124]
miR-155	SHIP1	NK cell activation, IFN-γ production	human mouse	[113,114]
miR-181a/b	NOTCH1	NK cell maturation and enhanced IFN-γ production	human	[108]
miR-182	NKG2D NKG2A	Upregulation of NKG2D and Perforin-1, downregulation of NKG2A	human	[121]
miR-483-3p	IGF-1	NK cell development and cytotoxicity	human	[125]
miR-519a-3p	NKG2D ligands ULBP2 MICA	Impaired tumor cell recognition, NK activation, resistance toward granzyme B	human	[118]
miR-583	IL2Rγ	NK cell differentiation	human	[126]
miR-615-5p	IGF-1 R	Decreased CD56^dim^, increased CD56^bright^ NK cell subsets and reduced the cytotoxic markers NKG2D, TNF-α and perforins	human	[119]

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
