# Peer review of "Natural Killer Cells as Key Players of Tumor Progression and Angiogenesis: Old and Novel Tools to Divert Their Pro-Tumor Activities into Potent Anti-Tumor Effects"

_cancers, 2019, doi:10.3390/cancers11040461_

Round 1
Reviewer 1 Report
Although the manuscript has been somewhat improved, significant problems remain such that, in my opinion, it is still not suitable for publication.
I am satisfied that the authors have now integrated a discussion of the important and relevant paper by Gao (2017, Nat Immunol) into the effects of TGFb on NK cells in cancer. I am also satisfied by the more up-to-date treatment of NK cells in immunotherapy. The final paragraph is improved in that it no longer overstates the novelty of the idea of targeting NK cells in cancer.
Much of the referencing is improved. However, I still notice a couple of the rogue references that I called attention to in my earlier review. Reference 114, which ought to point to a paper about a mouse knockout of miR29b still points to a paper on the predictive significance of miR29b in AML patients. Reference 139, which ought to point to point to a paper on neutrophil interactions with other leukocytes still points to a paper on the identification of a novel pathogen of fruit flies
Now that the section on HLA-G is properly referenced I can better assess it. One thing I note is that the authors have not discussed the controversy around the expression of HLA-G by tumours. In particular, some commonly used monoclonal antibodies to HLA-G give false positive results (Apps, 2008, Trends Immunol is an old review, but useful). I feel that the authors ought at least to mention this, so that the reader can approach the positive studies that they cite with an appropriate degree of caution.
Finally, the authors continue to put forward the out-of-date view that NK cells are ILC1 (at line 57), despite citing the paper in which it is proposed that they should now be considered a separate lineage (Vivier, 2018, Cell). It is arguable, particularly in the context of this review, that NK cells can transition to ILC1 in the TME and that the distinction between NK and ILC1 is therefore somewhat artificial. However, to make this nuanced argument, the authors first have to start by using the current definitions of the various cell types and explicitly show how they differ in the TME, rather than by ignoring current thinking.
Author Response
Comments and Suggestions for Authors
Although the manuscript has been somewhat improved, significant problems remain such that, in my opinion, it is still not suitable for publication.
I am satisfied that the authors have now integrated a discussion of the important and relevant paper by Gao (2017, Nat Immunol) into the effects of TGFb on NK cells in cancer. I am also satisfied by the more up-to-date treatment of NK cells in immunotherapy. The final paragraph is improved in that it no longer overstates the novelty of the idea of targeting NK cells in cancer.
We thank the reviewer for the positive comments.
Much of the referencing is improved. However, I still notice a couple of the rogue references that I called attention to in my earlier review. Reference 114, which ought to point to a paper about a mouse knockout of miR29b still points to a paper on the predictive significance of miR29b in AML patients. Reference 139, which ought to point to point to a paper on neutrophil interactions with other leukocytes still points to a paper on the identification of a novel pathogen of fruit flies.
We apologize for the problem with references that have now been fixed.
Now that the section on HLA-G is properly referenced I can better assess it. One thing I note is that the authors have not discussed the controversy around the expression of HLA-G by tumours. In particular, some commonly used monoclonal antibodies to HLA-G give false positive results (Apps, 2008, Trends Immunol is an old review, but useful). I feel that the authors ought at least to mention this, so that the reader can approach the positive studies that they cite with an appropriate degree of caution.
We discussed the controversy around the expression of HLA-G by tumours and provided related citation. This will give a more appropriate view on the studies concerning HLA-G and cancer.
Finally, the authors continue to put forward the out-of-date view that NK cells are ILC1 (at line 57), despite citing the paper in which it is proposed that they should now be considered a separate lineage (Vivier, 2018, Cell). It is arguable, particularly in the context of this review, that NK cells can transition to ILC1 in the TME and that the distinction between NK and ILC1 is therefore somewhat artificial. However, to make this nuanced argument, the authors first have to start by using the current definitions of the various cell types and explicitly show how they differ in the TME, rather than by ignoring current thinking.
We have followed the comment of changing the text from lines 57 to 64 clearly indicating that classical NK cells are different from ILC based on the current definition. However, we think that it is really difficult in human beings to characterize phenotypically and functionally ILCs due to the very low number of these cells in tissues and the lack of surface expression of specific markers.
Reviewer 2 Report
The manuscript submitted by Bassani B. et al aims to describe the proangiogenic and protumoral function of NK cells. The work is very exhaustive and generally well written but in some sections (i.e. from section 2.1.3 to 2.1.6) it seems not so close to the purpose indicated, digressing more about the tumor-associated mechanisms that suppress NK cells cytotoxicity rather than stressing an active regulatory role of NK cells on angiogenesis and tumor progression. Along this line paper might be improved by shortening some sections or briefly integrating those in the other sections of the manuscript, keeping the focus on the proangiogenic and protumoral aspects. The complexity of the review is paralleled by figures that, although graphically good, hold too much information, difficult to appreciate. It follows that figure legends are too long. To conclude, simplifying figures and changing text according to suggestions might improve the quality of the manuscript.
Minor:
Line 59-60: “Indeed...ILC1”....sentence is not clear
Line 203: “that that”
The following references might be added and discussed where appropriate. Reference 2 should be added in the table
1) Casu B. et al, Cancers 2019 “Novel Immunoregulatory Functions of IL-18, an Accomplice of TGF-β1.”
2) Regis S. et al., Frontiers in Immunology 2017 TGF-β1 Downregulates the Expression of CX3CR1 by Inducing miR-27a-5p in Primary Human NK Cells.
Author Response
Comments and Suggestions for Authors
The manuscript submitted by Bassani B. et al aims to describe the proangiogenic and protumoral function of NK cells.
This comment is partially correct. We aim at discussing how NK cells can support tumor progression and how induction of angiogenesis, due to TME stimuli, can be a relevant part on the associated tumor supporting activities. To make this aim clearer we modified the title as “Natural killer cells as key players of tumor progression and angiogenesis: old and novel tools to divert their pro-tumor activities into potent anti-tumor effects. The abstract has been modified, accordingly.
The work is very exhaustive and generally well written but, in some sections, (i.e. from section 2.1.3 to 2.1.6) it seems not so close to the purpose indicated, digressing more about the tumor-associated mechanisms that suppress NK cells cytotoxicity rather than stressing an active regulatory role of NK cells on angiogenesis and tumor progression. Along this line paper might be improved by shortening some sections or briefly integrating those in the other sections of the manuscript, keeping the focus on the proangiogenic and protumoral aspects.
We agree with the reviewer. For section 2.1.3, we integrate by proposing a dual role of PGE2 in supporting tumor angiogenesis. The direct and recognized role of PGE2 di per sè, and those mediated by the induction of anergic NK cells that, at least in solid tumors, has been demonstrated as a crucial step to induce the angiogenic switch on NK cells.
Section 2.1.6 cannot be deleted or shortened. Therefore, we decided to integrate the miRs section with that of EVs, that go often paired, according to the literature.
The complexity of the review is paralleled by figures that, although graphically good, hold too much information, difficult to appreciate. It follows that figure legends are too long. To conclude, simplifying figures and changing text according to suggestions might improve the quality of the manuscript.
We feel that simplify figure will “sacrifice” relevant concepts. Therefore, we divided complex figures and provided concise legends. This will help the reader to not miss relevant concept and trace them in less complex figures.
Minor:
Line 59-60: “Indeed...ILC1”....sentence is not clear
Line 203: “that that”
The following references might be added and discussed where appropriate. Reference 2 should be added in the table
Casu B. et al, Cancers 2019 “Novel Immunoregulatory Functions of IL-18, an Accomplice of TGF-β1.”
Regis S. et al., Frontiers in Immunology 2017 TGF-β1 Downregulates the Expression of CX3CR1 by Inducing miR-27a-5p in Primary Human NK Cells.
Reference has been integrated and commented in the TGFb section.
Round 2
Reviewer 1 Report
I am now satisfied that this manuscript is suitable for publication.
This manuscript is a resubmission of an earlier submission. The following is a list of the peer review reports and author responses from that submission.
Round 1
Reviewer 1 Report
In this work the authors discuss the possibility that cancer environment can induce alterations in NK cell phenotype and function that could be contributing to cancer progression. They also review novel therapeutic approaches aiming to potentiate NK cell activities against tumours.
The manuscript is an extensive review, that is well written and clearly organised. The literature quoted (more than 190 references) is pertinent and updated, with many references published in 2017-2018. The figures are well designed, clear and self-explanatory.
Minor comment.
The authors should consider rewriting the abstract and the conclusions to clarify their final message.
Author Response
According to the reviewer 1 in this new version we have checked all the English text and modified and improved in particular both the Abstract and the Conclusions.
We have also corrected the errors of some references and added new recent references of interest.
We hope that this new version now has the qualities to be accepted for the publication.
Reviewer 2 Report
In this review, the authors set out to consider all aspects of the relevance of NK cells to cancer biology. Probably for this reason, the review as it currently stands lacks focus and fails to come to any useful conclusions about the importance of NK cells in cancer. Indeed, in the final paragraph, the authors propose that NK cells are “a relevant host-dependent hallmark of cancer and a new cellular paradigm in tumor progression”. Far from being new, this idea is now a couple of decades old.
There are a number of problems with the references, which is a fatal flaw in a literature review. For example, at line 81 “Kestin et al showed that TGF-B promotes conversion of…” has no reference and, since the authors have also misspelled the author’s name (they mean Keskin, 2007, PNAS) the interested reader would not be able to find the study. The references in the entire section on HLA-G point to papers on miRs (and no papers on HLA-G are present in the reference list) making this section useless. Reference 88, which ought to point to a paper about a mouse knockout of miR29b instead points to a paper on the predictive significance of miR29b in AML patients. Reference 111, which ought to point to point to a paper on neutrophil interactions with other leukocytes instead points to a paper on the identification of a novel pathogen of fruit flies.
In addition to these very basic errors, the literature reviewed is not up to date. For example, at line 49 the authors say that “recently NK cells have been classified as ILC1” (with no citation). This was the view put forward by Spits et al in 2013 (in Nat Rev Immunol). The 2018 view, put forward by Vivier et al in Cell is that NK cells form a lineage separate from ILC1. The authors also fail to consider the recent literature on how NK cells may be converted to ILC1 in tumors by TGFb (Gao, 2017, Nat Immunol). This is particularly surprising given that they dedicate an entire section to TGFb. Further, the field of NK immunotherapy for cancer is very fast-moving and exciting at the moment, but the authors fail to mention any of the recent findings in the section on this topic (for example, Bjorklund, 2018, Clin Cancer Res; Hermanson, 2018, Cell Stem Cell; Fleischhauer, 2018, Bone Marrow Transplant; Schaffer, 2016, Biol Blood Bone Marrow Transplant; Bachanova, 2014, Blood – not an exhaustive list).
A number of really excellent reviews on NK cells in cancer and cancer immunotherapy, which cite all the current literature, have recently been published. In its current form, this review does not add anything to the field.
Author Response
According to the reviewer 2 and 1 in this new version we have corrected and improved all the English text and modified several sections strengthening our thoughts and our observations concerning this field of investigation.
We partially agree with the reviewer’s comment. Our review has been focused on selected soluble and cellular TME components orchestrating NK cell contribution to cancer progression. “Old” actors are necessarily included since their pivotal importance in governing NK cell activity. At the same time, it’s exhaustively discussed how these soluble/cellular mediators are involved in a totally new concept (the authors are pioneers in this topic), that is the pro-angiogenic activities of NK cells in cancers. We further support our statement that “a relevant host-dependent hallmark of cancer and a new cellular paradigm in tumor progression”, stressing the new insights of these idea.
We apologize for the problems with the references, probably due to an incomplete running of End Note program. We have corrected and rechecked all the corrections. References now fit and are coherent with the referred text.
We apologize, once again, for having missed some pivotal reference article. The revised version now better discusses the findings by Vivier et al. on NK cell origin and we integrated the TGFb section with the insights regarding the ability of TGFb to convert NK cell into ILC1 in tumors.
We thank the reviewer for these arguments. We have reflected on these critical and negative comments and we think that now, in our new version, we have considerably improved our work both in the investigations and updates requested and in the clarity of table and the figures and respective legends. In particular, we integrated the section of NK cell immunotherapy with the suggested references and other recent ones, showing new relevant insight in this specific field. We have also corrected the errors of some references and added new recent references of interest.
We hope that this new version now has the qualities to be accepted for the publication.
Reviewer 3 Report
Summary -
The manuscript by Bassani et al provides an overview of our current understanding of the tumor associated factors contributing to dysfunctional Natural Killer (NK) cells in the context of cancer. The authors provide a brief summary of selected soluble factors (e.g. TGFb) as well as relevant cell populations which have been shown to impact NK cell function (e.g. Macrophages). Additional information about current strategies to restore/enhance NK cytotoxic activity in cancer patients is also presented.
Major concerns -
1. The authors have provided a reasonable overview of a very broad topic. However, a considerable number of errors in language usage are found throughout the manuscript making it challenging to read and interpret which will make the review of limited use to the target audience. These errors are not limited to simple spelling mistakes and the authors are strongly encouraged to have a the manuscript reviewed prior to resubmission. A representative example - p11 line 416, "peculiarly stimulates CD8+ T cells" does not make any sense.
2. Although it is challenging to provide a review that is both comprehensive in breadth and depth in some sections the authors have provided little more that statements of fact without context. In particular the section on miRNAs was challenging to understand the significance of each of the individual miRNAs. The figure provided very little help (see point 3).
3. The amount of detail made interpretation of the figures difficult. In particular, the details provided in Figure 2 may be better presented as a table where the target of each miRNA species can be more clearly defined (and reference noted).
4. The conclusion that the presented data supports a role for NK cells in promoting cancer is confusing, p14 line 521. The data presented clearly establish a role for the TME in creating NK dysfunction but this is not the same as a role for NK cells in promoting tumor growth as implied by this statement see point 1.
Author Response
1.We apologize for the numerous errors along the text. We have now corrected and rechecked all the corrections. Spelling and eventual concept mistakes have been checked and revised.
2. We agree that figure 3 illustrate lots of concepts and interactions, even if letters are provided to guide the interpretation of the figure. Therefore, we provided more detailed figure legends to support the readers to figure interpretation.
3. We agree with the reviewer 3. Our revised version provides a table for the miRNA section that more clearly illustrate the concept presented in former Figure 2.
4. We rewrite the Conclusions to better elucidate the aim of our review, the insight provided, placing the TME/NK interactions, rather than NK cell alone as the new hallmark of cancers.
We hope that this new version now has the qualities to be accepted for the publication.